# Spatial variations and determinants of malnutrition among under-five children in Nigeria: A population-based cross-sectional study

**Lateef Babatunde Amusa** [1,2]*, **Waheed Babatunde Yahya**[1], **Annah Vimbai Bengesai**[3]

**1** Department of Statistics, University of Ilorin, Ilorin, Nigeria, **2** Centre for Applied Data Science, University of Johannesburg, Johannesburg, South Africa, **3** College of Law and Management Studies, University of KwaZulu-Natal, Durban, South Africa

* amusa.lb@unilorin.edu.ng

**Data Availability Statement:** All relevant data are within the paper and its Supporting information files.

## Abstract

Childhood undernutrition is a major public health challenge in sub-Saharan Africa, particularly Nigeria. Determinants of child malnutrition may have substantial spatial heterogeneity. Failure to account for these small area spatial variations may cause child malnutrition intervention programs and policies to exclude some sub-populations and reduce the effectiveness of such interventions. This study uses the Composite Index of Anthropometric Failure (CIAF) and a geo-additive regression model to investigate Nigeria's prevalence and risk factors of childhood undernutrition. The geo-additive model permits a flexible, joint estimation of linear, non-linear, and spatial effects of some risk factors on the nutritional status of under-five children in Nigeria. We draw on data from the most recent Nigeria Demographic and Health Survey (2018). While the socioeconomic and environmental determinants generally support literature findings, distinct spatial patterns were observed. In particular, we found CIAF hotspots in the northwestern and northeastern districts. Some child-related factors (Male gender: OR = 1.315; 95% Credible Interval (CrI): 1.205, 1.437) and having diarrhoea: OR = 1.256; 95% CrI: 1.098, 1.431) were associated with higher odds of CIAF. Regarding household and maternal characteristics, media exposure was associated with lower odds of CIAF (OR = 0.858; 95% CrI: 0.777, 0.946). Obese maternal BMI was associated with lower odds of CIAF (OR = 0.691; 95% CrI: 0.621, 0.772), whereas, mothers classified as thin were associated with higher odds of CIAF (OR = 1.216; 95% CrI: 1.055, 1.411). Anthropometric failure is highly prevalent in Nigeria and spatially distributed. Therefore, localised interventions that aim to improve the nutritional status of under-five children should be considered to avoid the under-coverage of the regions that deserve more attention.

## Introduction

Malnutrition, a pathological state resulting from deficiencies (or excesses) in the intake of energy, protein of micronutrients, is a serious public health problem accounting for more than

**Funding:** The author(s) received no specific funding for this work.

**Competing interests:** The authors have declared that no competing interests exist.

40% of under-five children globally. Despite some achievements and partial success, progress in many countries has been slow, especially in sub-Saharan Africa. About 150 million (21.9%) under-five children are stunted, while 49.5 million (7.3%) are wasted. The situation is worse in Africa, where the prevalence of stunting is approximately 29%, which is higher than the global average of 21%. In comparison, at least 7% of under-fives are wasted [1]. Overweight rates have also increased by almost 70%, from 6.6 million in 2000 to 9.7 million in 2017 [1].

The burden of malnutrition is also of significant proportions in Nigeria. The 2018 Nigeria Demographic and Health Survey (NDHS) reported that stunting prevalence in children up to 60 months was 37%, with 17% severely stunted, 29% either underweight or severely underweight, and 9% were either wasted or severely wasted [2]. Similar statistics were reported in the 2018 National Nutrition And Health Survey (NNHS) [3]. All these studies suggest that under-five malnutrition is a persistent issue of epidemic proportions [4]; hence more studies need to investigate its magnitude and epidemiology.

While the three basic anthropometric indices (stunting, wasting, & underweight) of malnutrition have been used widely to evaluate childhood nutritional status, they do not provide a complete picture of the overall burden of malnutrition [5, 6]. This is because each of these indices reflects distinct biological processes, which are not mutually exclusive and often co-occur. For instance, stunting that leads to low height for the age indicates long-term undernutrition, while wasting (low weight for height) results from acute undernutrition. At the same time, being underweight is a composite of acute and chronic undernutrition [7]. Therefore, either of these indices might not sufficiently capture a child's actual and overall malnutritional status.

Accordingly, the Composite Index of Anthropometric Failure (CIAF), an aggregate measure of individual malnutritional indicators, was proposed [8]. The CIAF has been used successfully in studies from India [9, 10], China [11], Malawi [12], and recently, Ethiopia [6]. However, such studies are yet to be conducted in Nigeria, despite having a high malnutrition burden. Thus, available statistics might underestimate the overall magnitude of malnutrition. Furthermore, most of the existing studies have examined spatial variations of CIAF at macro levels such as country or region [6], with less focus being placed on zonal (or state) level variation [6]. However, for a country like Nigeria, which has a federal system where administrative, fiscal and political systems are decentralised, examining the issue of malnutrition and its variations across administrative zones will lead to a nuanced understanding of the healthcare needs of young children.

This study examines the prevalence of anthropometric failure in Nigeria using the aggregate CIAF index. Using a geo-additive regression model, we sought to investigate spatial variation in the burden of under-five malnutrition and determine its socio-demographic and environmental determinants at the parental, child, household, and community levels.

## Material and methods

### Data description and management

Data were extracted from the child and household files from the 5$^{th}$ round (2018) of the Nigeria Demographic Health Survey (NDHS). The National Population Commission (NPC) conducted the survey in partnership with the Federal Ministry of Health and with technical assistance from the DHS Program (www.measuredhs.com) [2]. NDHS data is a nationally representative sample that provides information on fertility and family planning, as well as demographic and health indicators, for women and men between the ages of 15 and 49. The survey further provides information about important aspects of child and infant health, including neonatal and post-neonatal care, infant deaths, child feeding practices, and child nutritional status.

**Table 1. CIAF classification of nutritional status in under-five children.**

| Group | Description | Stunting | Wasting | Underweight |
|---|---|---|---|---|
| A | No failure | No | No | No |
| B | Stunting only | Yes | No | No |
| C | Stunting + underweight | Yes | No | Yes |
| D | Wasting + underweight + stunting | Yes | Yes | Yes |
| E | Wasting + underweight | No | Yes | Yes |
| F | Wasting only | No | Yes | No |
| G | Underweight only | No | No | Yes |

**Ethical Approval.** The survey got an ethical clearance from the ethical committee of ICF Macro (Calverton, MD, USA). Details of this data have been described elsewhere [2, 13, 14]. The CIAF is the outcome variable of this study. This index, as classified in Table 1, is calculated from three standard anthropometric measures of nutritional status among under-five children: height-for-age (stunting), weight-for-height (wasting), and weight-for-age (underweight). These measurements were standardised via the construction of Z-scores, normalised by the World Health Organization's 2006 Child Growth Standards [15, 16]. Details about these anthropometric measures can be found in the DHS final report [2].

Further, the WHO-recommended cut-offs were used to define the three types of malnutritional status in children (Stunting, wasting, and underweight), which were respectively classified based on their z-scores relative to the median value of a healthy reference population. Accordingly, a child is classified as stunted, wasted, and underweight if their respective z-scores are less than two standard deviations: height-for-age (HAZ < -2), weight-for-height (WHZ < -2), and underweight (WAZ < -2). The three measures were then classified binarily (stunted vs. non-stunted, wasted vs. non-wasted, and underweight vs. non-underweight) for subsequent calculation of the CIAF [17–21]. Computation of the CIAF requires that the children's nutritional indicators are divided into seven (A-G) categories. A child is thus considered malnourished, as measured by the CIAF, if the child suffers any of the anthropometric failure (B-G), as shown in Table 1 [5].

The child, household, and maternal covariates were chosen based on literature results and available data as potential risk factors [18, 22–24]. For instance, at the parental level, factors such as maternal age, education, and health have been studied as potential confounders of child undernutrition [25–27]. Child-level factors include breastfeeding status [28, 29], ill-health [30], birth order [31], and birth weight [32]. There is also evidence that children from poor households suffer more anthropometric failures than wealthy families [33]. Household-level socioeconomic status has also been associated with child malnutrition [34, 35].

A description of the selected variables can be found in Table 2. The respondents' geographic information was extracted as a boundary file using the DHS-provided administrative shape files, which were then recoded to match the 36 states and the Federal Capital Territory (FCT).

## Descriptive statistics

Our eligible sample comprised 11,694 under-five children whose mothers were interviewed for complete information about their households. However, due to missing values on the study variables and removing children with invalid anthropometric measurements, the analytic sample was reduced to 10,962.

**Table 2. The description of all the variables in the 2018 NDHS data.**

| Variables | Description |
|---|---|
| Age (0–59 months) | Age of child |
| Mother's age (15–49 years) | Age of mother |
| Diarrhoea (Yes, No*) | Presence of diarrhoea two weeks before the survey |
| Cough (Yes, No*) | Presence of cough two weeks before the survey |
| Fever (Yes, No*) | Presence of fever two weeks before the survey |
| Vitamin A received (Yes, No*) | Received vitamin A six months before the survey |
| Birth order (1st birth*, 2nd-3rd, 4th or higher) | Child's position of birth among siblings |
| Gender (male, female*) | Gender of child |
| Size at birth (very small*, small, average or larger) | Reported size of the child at birth |
| Breastfeeding (Yes, No*) | The child is currently breastfeeding |
| Education (secondary & higher, primary, none*) | Mother's highest educational level |
| BMI (Obese, thin, normal*) | Mother's body mass index |
| Working (Yes, No*) | Mother's working status |
| Toilet facility (improved, unimproved*) | Type of household toilet facility |
| Water facility (Improved, unimproved*) | Household water source |
| Residence (urban, rural*) | Household type of residence |
| Wealth index (richest, richer, middle, poorer, poorest*) | Household wealth index |
| Media exposure (Yes, No*) | Exposure to mass media (radio, television, and newspapers) at least once a week |

*Reference category

## Inferential statistics

**Geo-additive model.** We briefly describe the structured additive regression (STAR) or geo-additive model employed in our analysis. The models will additionally be described from a Bayesian perspective.

In modelling the CIAF status of the under-five children, we define $Y_{hi} \sim Bernoulli(1, \pi_{hi})$, where $\pi_{hi} = P(Y_{hi} = 1)$ is the probability that child $i$ and state $h$ is undernourished. Assume the matrix $(v_i, x_{ir}, s_h)$, where $v_i$ is a vector of categorical covariates; $x_{ir}$ is a vector of continuous covariates; $s_h$ is the child's state of residence at the time of the survey, the geo-additive model is thus given by

$$Logit = (\pi'_{hi}) = \eta_{hi} = v'_i\beta + \Sigma_{r=1}^{p} f_r(x_{ir}) + f_{spat}(s_h), \ i = 1, 2, \ \ldots, \ 10962; \ h = 1, 2, \ldots, 37 \ (1)$$

where $\beta$ is a vector of fixed effect parameters modeled linearly; $f_r$, $r = 1, 2, \ldots, p$ are non-linear smooth functions of the continuous covariates; $f_{spat}$ are the spatial effects of the state of residence $s_h$. The spatial effects $f_{spat}$ may be further divided into a spatially correlated (structured) and an uncorrelated (unstructured) effect as

$$f_{spat}(s_h) = f_{str}(s_h) + f_{unstr}(s_h) \tag{2}$$

The structured spatial effect $f_{str}(s_h)$ assumes that states which are close in proximity are more likely to be correlated and have strong similar spatial structures, whereas unstructured spatial effect $f_{unstr}(s_h)$ represents the spatial variation induced by unmeasured state-level factors that are not spatially related and are only present locally [36].

**Specification of Bayesian prior distributions and hyper-parameters.** As required for Bayesian inference, appropriate priors are assumed for the parameters and functions. Independent diffuse priors were assumed on the fixed effects parameters, and the Bayesian P-splines priors [37, 38] were assumed for the non-linear effects. The P-spline can be represented as a linear combination of the basis function (B-spline): $f(x) = \Sigma_{t=1}^{r} \beta_t\, B_t(x)$, where $B_t(x)$ are B-spline basis functions and the coefficient $\beta_t$ are unknown regression coefficients assigned to follow a 1$^{st}$ order $\beta_t = \beta_{t-1} + U_t$, or a 2$^{nd}$ order $\beta_t = 2\beta_{t-1} - \beta_{t-2} + U_t$ Gaussian random walk priors. The error term $U_t$ are i.i.d. (independent and identically distributed) errors $\sim N(0, \chi^2)$. If 1$^{st}$ order is assumed, diffuse prior $\beta_{r1}$ will be chosen as constants for initial values, while diffuse priors $\beta_{r1}$ and $\beta_{r2}$ are chosen if 2$^{nd}$ order is assumed. The smoothness of the function $f_r$ is controlled by the variance component $\chi^2$. It has a weakly informative and widely dispersed inverse gamma (IG) prior, which is obtained by selecting very small hyperparameters a and b. Though, standard choices of hyperparameters a and b are 1 and 0.005 or a = b = 0.001 [39–41].

For the spatial effects, a Gaussian Markov random field (GMRF) prior [42] was assumed. Two states *s* and *t* are regarded as neighbours if they share a common boundary.

$$\text{GMRF} : \left(f_{spat}(s)/f_{spat}(t);\ t \neq s, \chi^2_{str}\right) \sim N\left(\Sigma_{t\varepsilon\delta_s} f_{str}(t)/N_s,\ \chi^2_{str}/N_s\right) \tag{3}$$

where $N_s$ is the number of neighbors of state *s*, $\delta_s$ are the neighbours of state *s*, and $t\ \varepsilon\ \delta_s$ denotes that state *s* is a neighbour of state *t*. The smoothness of the spatial effect is controlled by the variance $\chi^2_{str}$ and accounts for spatial variation between the states. The unstructured spatial effect $f_{unstr}(s)$ will be assigned i.i.d. gaussian priors and specified as

$$f_{unstr}(s) \sim N \sim \left(0,\ \frac{1}{\chi^2_{unstr}}\right) \tag{4}$$

## Analysis

We used second-order P-splines for the non-linear effect parameters, diffuse priors for the fixed effects parameters, and a Gaussian Markov random field prior for the spatial effects parameters. The typical selection of a = b = 0.001 were used for the variance parameters. A sensitivity analysis was performed where different hyperparameter values were examined for a significant change in results [43]. However, the estimates did not suggest substantial differences from the standard hyperparameter values. We conducted 12000 MCMC iterations, thinning out the Markov chain every 10th iteration, resulting in a random sample size of 2000 for each parameter.

## Model selection

We compared the model fit statistics of four competing models with varying degrees of complexity. The models will be hierarchically fitted as follows:

$$\text{M0}:\ Logit\,(\pi'_{hi}) = v'_i\beta \,(\text{Linear fixed effects for all variables excluding spatial effects}) \tag{5}$$

$$\text{M1}:\ Logit\,(\pi'_{hi}) = v'_i\beta + f_1(x_{i1}) + f_2(x_{i2}) + \ldots f_p\left(x_{ip}\right)\,(\text{Linear fixed effects + non-linear}$$
$$\text{effects of the continuous variables}) \tag{6}$$

$$\text{M2}:\ Logit\,(\pi'_{hi}) = v'_i\beta + f_{spat}(s_h)\,(\text{Linear fixed effects for all variables + spatial effects}) \tag{7}$$

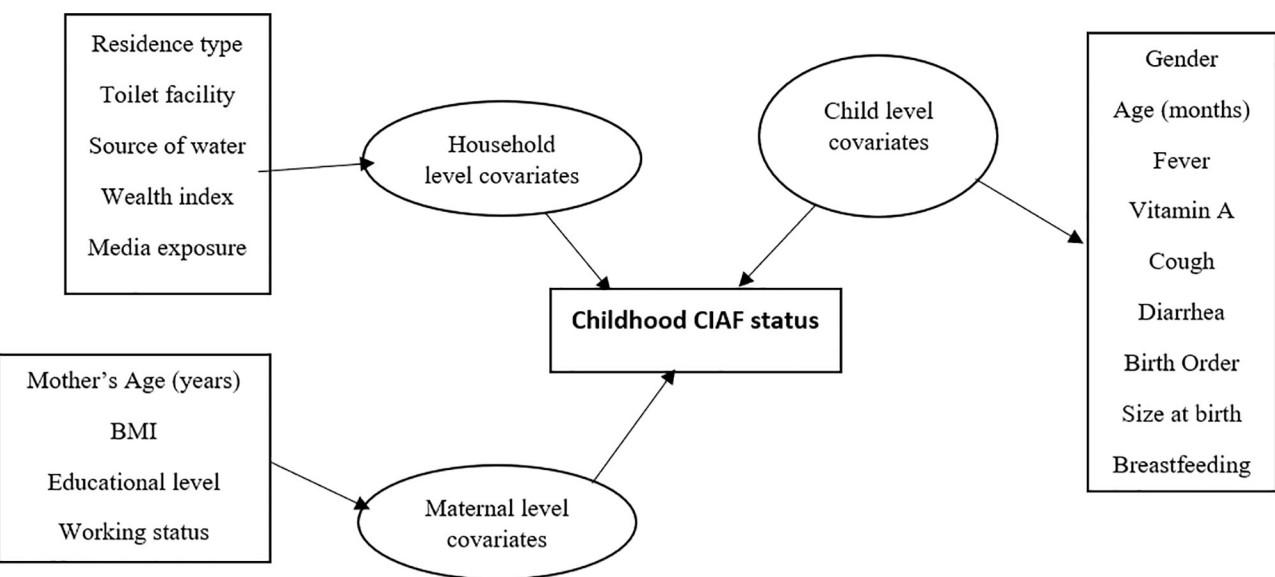

**Fig 1. Potential risk factors of CIAF considered in the geoadditive regression model.**

$$M3 \text{ (full model)}: \; Logit\,(\pi'_{hi}) = v'_i\beta + \Sigma_{r=1}^p f_r(x_{ir}) + f_{spat}(s_h) \text{(Linear fixed effects + non-linear effects}$$
$$\text{of the continuous variables + spatial effects)} \tag{8}$$

The best model choice is based on the deviance information criteria (DIC) [44] and its corresponding effective number of parameters pD. The DIC is computed as IC = pD + $\bar{D}$, where the effective number of parameters is pD (model complexity measure), and $\bar{D}$ is the posterior mean of the model deviance. Since small DIC values indicate a better fit, the final model will be selected based on producing the smallest DIC value.

We included the categorical predictors as linear fixed effects. The two continuous variables, child's and mother's ages, were modeled as non-linear effects, while a spatial effect was used for the states of residence. The inference (parameter estimation) was done using the generic Markov Chain Monte Carlo (MCMC) sampling. The potential risk factors of CIAF, as considered in the geoadditive regression model are presented in Fig 1.

Data analysis was done using the R2BayesX [45] package (version 0.3–1) of the R statistical software [46].

Table 3 presents the results of the DIC and the effective number of parameters for each of the four fitted models. As evident from Table 4, the geo-additive model, M3, yielded the smallest DIC (pD) value and was thus chosen as the best model among the four models fitted. Therefore, the discussion of the results will be based on model M3.

**Table 3. Model's goodness of fit comparison and selection based on DIC, pD and deviance.**

|  | M0 | M1 | M2 | M3 |
|---|---|---|---|---|
| DIC | 13280.35 | 13023.87 | 12825.88 | **12710.03** |
| pD | 24.89 | 38.38 | 55.48 | **66.47** |
| Deviance | 13230.58 | 12947.11 | 12936.84 | **12577.1** |

**Table 4. Prevalence of the different forms of anthropometric failure of the children in the 2018 NDHS data.**

| Anthropometric status | Prevalence (95% Confidence Interval) |
| --- | --- |
| CIAF | 41.3% (40.4%, 42.2%) |
| No failure | 58.7% (57.8%, 59.6%) |
| Wasting Only | 6.8% (6.3%, 7.2%) |
| Stunting Only | 36.2% (35.3%, 37.0%) |
| Underweight only | 21.7% (21.0%, 22.5%) |
| Wasting + Underweight | 2.1% (1.8%, 2.3%) |
| Stunting + Underweight | 14.8% (14.2%, 15.5%) |
| Wasting + Stunting + Underweight | 3.0% (2.7%, 3.4%) |

## Results

### Factors associated with outcome

This study sought to determine the risk factors and spatial patterns of CIAF among under-five children in Nigeria using the 2018 NDHS data. The posterior odds ratios and associated 95% credible intervals are reported for the linear fixed effects. For the non-linear and spatial effects, the posterior means (log-odds) and associated 95% credible intervals were presented. Accordingly, fixed effects are considered significant if the 95% credible interval does not contain 1, whereas the credible intervals for the non-linear and spatial effects should not contain zero. Positive and negative coefficient estimates correspond to increased and decreased CIAF risk, respectively.

As shown in Table 4, CIAF identified more children (41.3%; 95% CI: 40.4%, 42.2%) having anthropometric failure than the conventional indices, including stunting, wasting, and underweight. The results further suggest that among the individual index that makes up the CIAF, stunting (36.2%; 95% CI: 35.3%, 37.0%) is more prevalent. In terms of geographical variation, there is an apparent north-south disparity in the CIAF prevalence, as shown in Fig 2. For instance, a northern state, Sokoto, was the most CIAF prevalent state at 68.1% (95% CI: 60.0, 68.2), whereas a southern state, Anambra, was the least CIAF prevalent state at 17% (95% CI: 15.9, 17.4).

### Fixed effects

Tables 5 and 6 present the results of the linear fixed effects, which show the posterior odds ratios and 95% credible intervals. Compared to their counterparts, significantly higher odds of CIAF were suggested for male children, those who had diarrhoea two weeks before the survey, those who were being breastfed, and children in the fourth or higher birth order. On the other hand, children reported to be average/larger or small at birth were associated with significantly lower odds of CIAF compared to those reported to be very small (Table 5).

As for the household-level factors, Table 6 showed that a few variables were significant. The odds of CIAF were less likely for children from households with media exposure and equally less likely for the richer wealth quintile (richer and richest) than those from the poorest families. Regarding maternal-related factors, children of mothers with primary and at least secondary education were associated with significantly lower odds of CIAF than those without education. The odds of CIAF were less likely for children whose mothers' BMI was obese, whereas those with thin mothers had higher odds of CIAF.

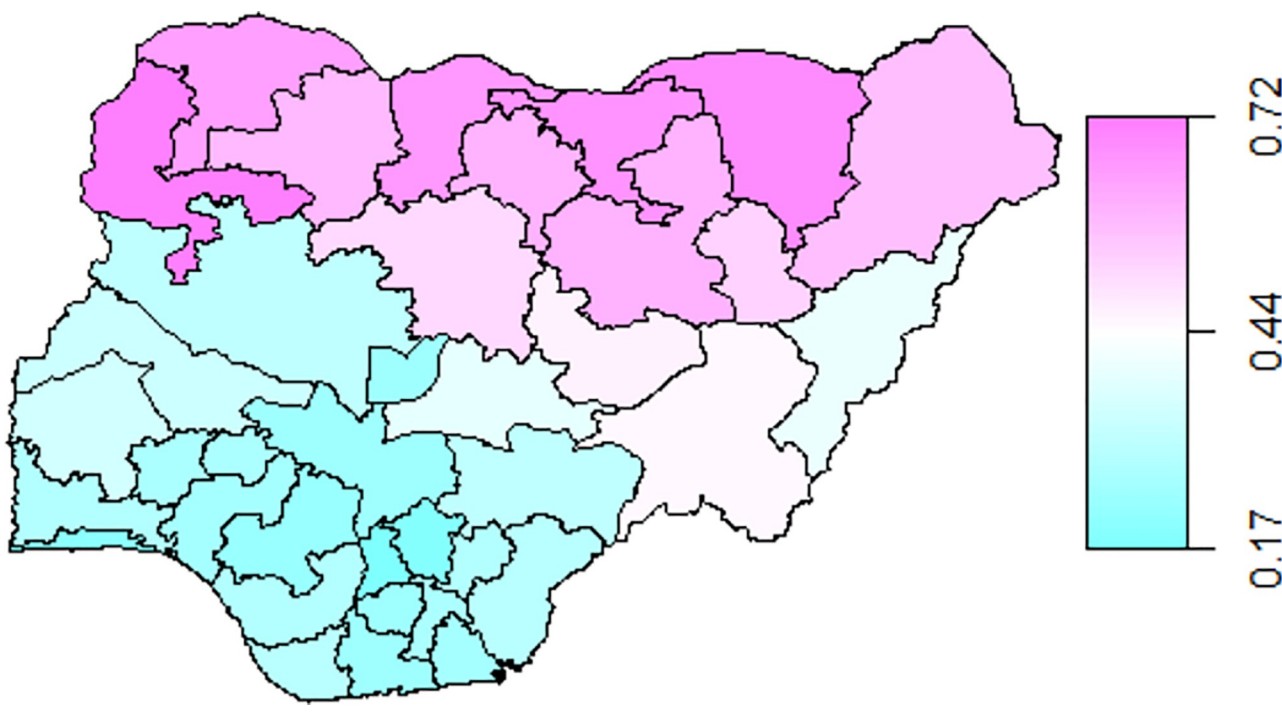

**Fig 2. The CIAF prevalence per Nigerian state based on NDHS 2018 data.**

## Non-linear effects

The estimates of the smooth term variance components suggest a linear effect for mother's age. Child's age, on the other hand, suggests a non-linear effect (Table 7). As shown in Fig 3, the log-odds of CIAF in under-five children were higher at the younger ages of their mothers and decreased as they grew older (Fig 3, Top panel). An Inverse U relationship was exhibited for the child's age. There was an increase in the effect of child's age till around 30 months, a decrease between 30 and 48 months, after which the effect increased beyond 48 months (Fig 3, Bottom panel).

**Table 5. Posterior estimates of the child-level fixed effects.**

| Factors | Posterior Odds Ratio | 95% Credible Interval | |
|---|---|---|---|
| Being breastfed (Yes vs No) | 1.128* | 1.016 | 1.250 |
| Size (small vs very small) | 0.673* | 0.507 | 0.883 |
| Size (average/larger vs very small) | 0.436* | 0.332 | 0.564 |
| Gender (male vs female) | 1.315* | 1.205 | 1.437 |
| Birth order (2nd or 3rd vs 1st) | 1.124 | 0.987 | 1.281 |
| Birth order (4th or higher vs 1st) | 1.454* | 1.234 | 1.706 |
| Vitamin A received (Yes vs No) | 0.983 | 0.890 | 1.089 |
| Fever (Yes vs No) | 1.018 | 0.912 | 1.137 |
| Cough (Yes vs No) | 0.979 | 0.866 | 1.109 |
| Diarrhoea (Yes vs No) | 1.256* | 1.098 | 1.431 |

NB: The asterisked posterior odds ratios are significant at the 5% level.

**Table 6. Posterior estimates of the maternal and household level fixed effects.**

| Factors | Posterior Odds Ratio | 95% Credible Interval | |
|---|---|---|---|
| Wealth index (poorer) | 0.941 | 0.820 | 1.080 |
| Wealth index (middle) | 0.889 | 0.766 | 1.038 |
| Wealth index (richer) | 0.672* | 0.554 | 0.811 |
| Wealth index (richest) | 0.515* | 0.409 | 0.651 |
| Working (Yes) | 1.063 | 0.962 | 1.179 |
| BMI (thin) | 1.216* | 1.055 | 1.411 |
| BMI (obese) | 0.691* | 0.621 | 0.772 |
| Residence (urban) | 0.948 | 0.850 | 1.064 |
| Education (primary) | 0.801* | 0.698 | 0.913 |
| Education (secondary & higher) | 0.622* | 0.539 | 0.719 |
| Water (Improved) | 0.941 | 0.851 | 1.046 |
| Toilet (improved) | 1.109 | 0.999 | 1.235 |
| Media exposure (Yes) | 0.858* | 0.777 | 0.946 |

NB: The asterisked posterior Odds ratios are significant at the 5% level.

## Spatial effects

Fig 4 displays the structured spatial effects on the log-odds of CIAF and their 95% credible intervals. Red (blue) shadings of states in the map of 95% credible interval indicate significantly higher (lower) estimates, whereas gray-shaded states are not significant. Results are not shown for the unstructured spatial effects as the estimates do not significantly differ from zero (Table 6).

In agreement with the descriptive results, a north-south divide exists in the level of CIAF across the Nigerian states, with substantial domination in most northern states. All the northwestern states (see Fig 4) are associated with a higher likelihood of CIAF. Other states with higher odds of CIAF include northeastern states (see Fig 4), namely Borno, Yobe, and Bauchi. Lower odds of CIAF are associated with Akwaibom, Imo, Anambra, Enugu, Ebonyi, Ondo, and Edo (southern parts), including only four north-central states: Niger, FCT, Kogi, and Benue. Other states have non-significant effects.

## Discussion

This paper uses the CIAF measure and a hierarchical geo-additive model to identify spatial and environmental factors that place some sub-groups at high risk of malnutrition and simultaneous anthropometric failures. Our approach was based on the understanding that the

**Table 7. Posterior estimates of smooth term variance components.**

| Variable | mean | SD | 95% Credible Interval | |
|---|---|---|---|---|
| | | | 2.5% | 97.5% |
| *Non-linear effect* | | | | |
| Child's age | 0.054 | 0.078 | 0.003 | 0.281 |
| Mother's age | 0.003 | 0.004 | 0.000 | 0.012 |
| *Spatial effect* | | | | |
| Structured | 0.323 | 0.100 | 0.166 | 0.555 |
| Unstructured | 0.009 | 0.011 | 0.001 | 0.039 |

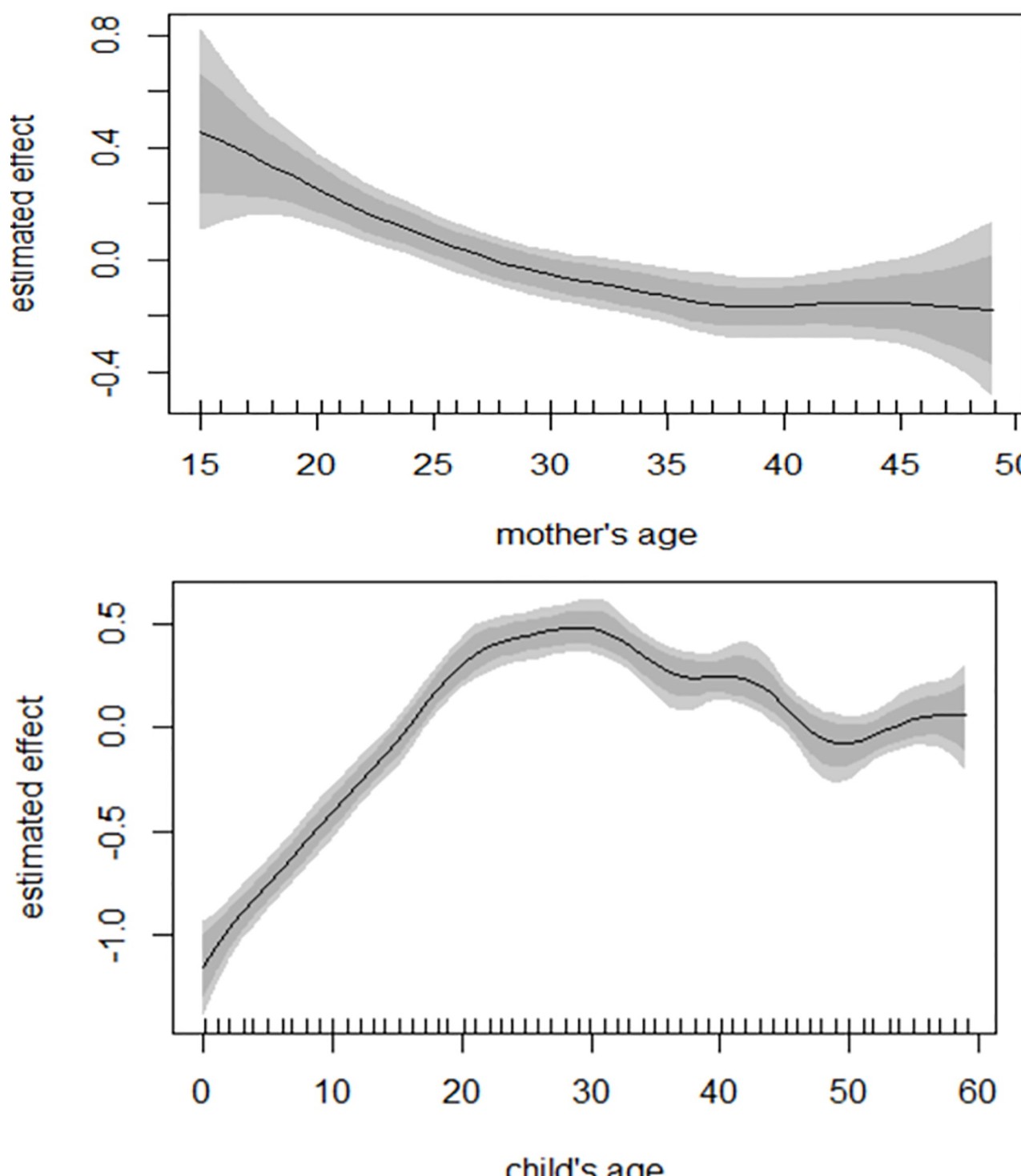

**Fig 3. Estimated non-linear effects of mother's age (Top panel) and child's age (Bottom panel).** Posterior means and their associated 95% credible intervals are shown.

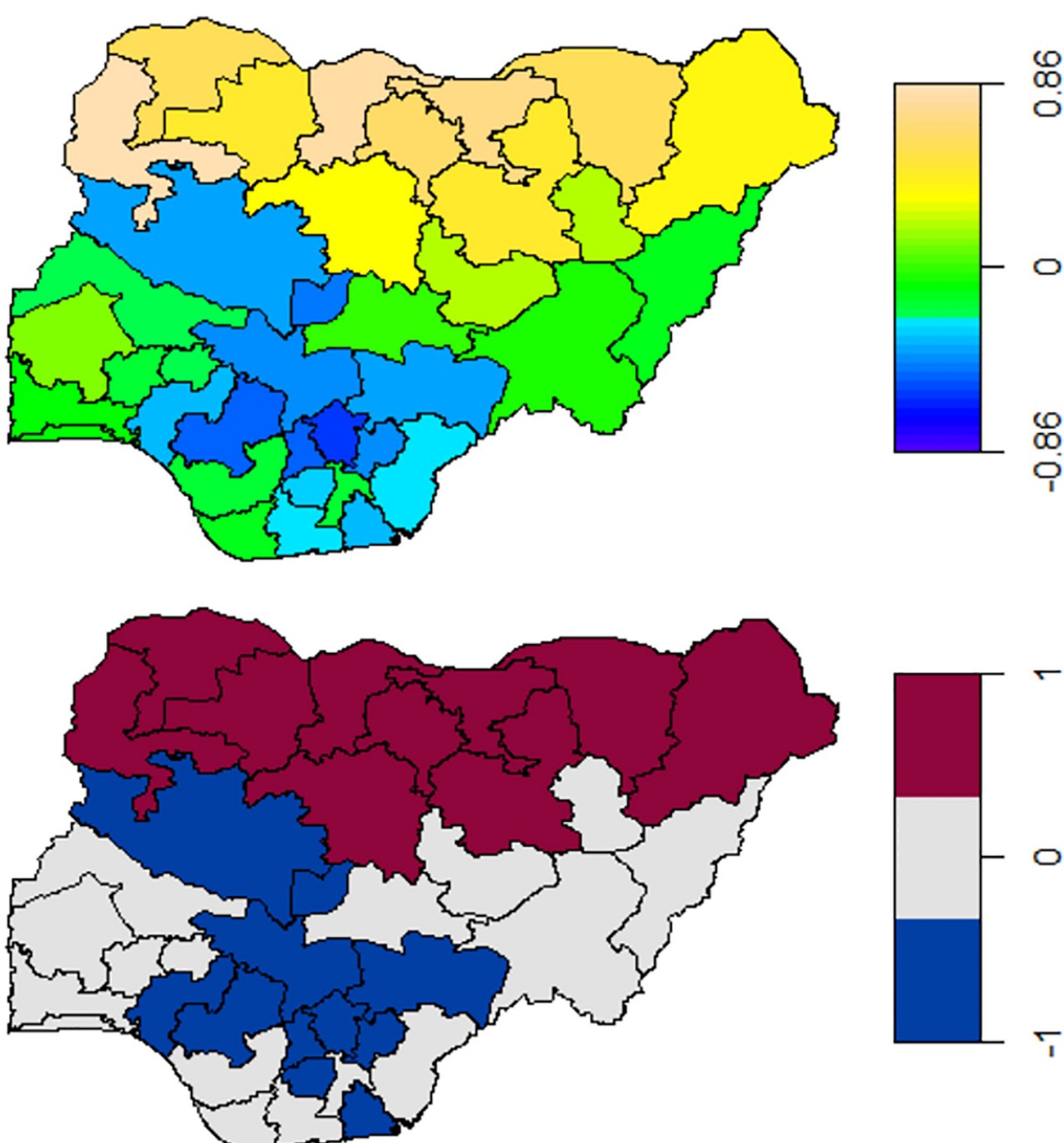

**Fig 4. The estimated posterior means of structured spatial effects (Top panel) and the 95% credible interval corresponding to the estimates of the structured spatial effects (Bottom panel).** NB: The scale on the map reads 1: significant positive effect, -1: significant negative effect, and 0: no effect.

interaction of individual and broader contextual factors which affect childhood undernutrition is not static across space and tends to vary geographically [47].

Overall, the malnutrition burden of Nigeria is evident in the prevalence of undernutrition, which using the CIAF index, was estimated at 41.3%. The descriptive maps also show

malnutrition hot spots, which confirm the north-south divide. Specifically, thirteen districts (seven northwestern and six northeastern states) were at higher risk of undernutrition. The northwestern were categorised as critical and with the highest risk. The multivariate analysis also confirmed these findings, where all the northwestern districts and three northeastern districts (except for Gombe, Adamawa, and Taraba) had significantly higher odds of undernutrition. In other words, our results suggest that geographical location is a significant determinant of child undernutrition, reflective of the spatial inequality in socioeconomic development in the regions [48]. There has been persistent poverty in Northern Nigeria. A recent study indicated that almost 50% of the poor live in the North-west, while the entire northern region accounts for 87% of poor Nigerians [49]. This clustering of undernutrition across CIAF-based hotspots confirms the need to focus on the country's poorer regions.

The role of parental education in children's developmental outcomes is well articulated in the literature [26, 50]. Therefore, it was not surprising that we found a significant association between our aggregate index of undernutrition and maternal education. A possible explanation for this finding is that educated mothers have access to information regarding better child care practices and hence can make informed decisions on children's dietary and health needs [51, 52]. Moreover, low parental education might also lead to low income, which amplifies the risk due to higher vulnerability to food insecurity. Related to this, we also found that household wealth significantly affected undernutrition. Again, this is likely due to insufficient dietary intake resulting from food insecurity.

Several studies have shown that access to media has a direct influence on health behaviours [53], including nutrition behaviours [54] in sub-Saharan Africa [55] and other South Asian countries [56], including Bangladesh [57]. Therefore, the study findings are consistent with previous studies. A plausible explanation for this association is that access to media also provides access to information, including information on nutrition and child health care. Media also has the advantage of simplifying complex scientific information, making it accessible to many women in a non-technical manner.

Poor nutrition during pregnancy can adversely affect outcomes like low birth weight—a risk factor for under-five malnutrition. Similar findings have been observed in other countries, such as Bangladesh [32], Malawi [33], and Pakistan [58]. The main explanation for this association is that children who have experienced intrauterine growth retardation are at risk of infant undernutrition. Even when these babies progress in life, they might not catch up on the lost growth while also experiencing other developmental problems [33, 59].

Birth order is one of the well-studied predictors of child undernutrition. Our study found that children of second, third, fourth, or higher birth order had a higher risk of anthropometric failure on the CIAF scale than those of first birth order. However, the results were only statistically significant for the latter. These findings mirror the results from other studies, such as in Bangladesh [31], India [60], and Africa as a whole [61], where higher birth order was associated with both stunting and wasting.

The observed effect of birth order on undernutrition might result from an alternative mechanism [62], such as family size, which can influence the nutritional status of all family members [63], including young children at risk of anthropometric failure. This is because the larger the family, the less resources available for each child, thus hindering physical and social development [64]. Hence, children in higher birth order are more likely to suffer because they need these nutrients for optimum growth. Thus, while birth order cannot be changed, understanding its impact on child developmental outcomes can be used to develop interventions that promote the healthy development of high-order children.

Diarrhoea is a well-known risk factor for malnutrition in under-five children and is associated with poor hygiene, access to clean water, and sanitation. While the link between diarrhoea

and nutrition can be explained through the loss of fluids and electrolytes and malabsorption of proteins and carbohydrates [30], some studies have also found positive associations between diarrhoea and family size, child sex, parental education, and household income [65, 66]. Thus, the association between the two is mediated by these family and socioeconomic conditions.

Counterintuitively, higher odds of CIAF were found among children currently being breastfed. Given that breastmilk has all the nutrients needed for adequate growth and development and prevents pathogen invasion [67, 68], we expected to find a negative relationship. While these unexpected findings could be attributed to uncontrolled confounding, some studies have shown that prolonged breastfeeding, i.e., beyond the first year of life, might be associated with undernutrition due to reverse causality [29]. In other words, it is not breastfeeding per se which causes undernutrition, but rather that delayed growth might influence mothers to continue breastfeeding [69]. Plausible as these explanations might be, they remain speculative given the nature of our data and perhaps an area for further research.

The present study is not without limitations. First, we drew our data from a cross-sectional design, making it difficult to investigate causal effects. Thus, only associations were estimated. Second, although the methods used in this study enable us to assess the overall prevalence of undernutrition in Nigeria, our analytical approach does not control for residual confounding. Indeed, other factors could be associated with undernutrition beyond what was captured in this study.

## Conclusion

This study has confirmed the spatial nature of undernutrition in Nigeria and established that both individual and household factors are essential in explaining regional variations in anthropometric failure in Nigeria. Therefore, interventions that aim to improve the nutritional status of under-five children should consider regions with a high prevalence of undernutrition to prevent the under-coverage of the regions or states that deserve more attention. All of these would assist tremendously towards attaining the Sustainable Development Goal (SDG) by improving the health conditions of under-five children in Nigeria.

## Supporting information

**S1 Data.**
(CSV)

## Acknowledgments

The authors would like to thank the DHS programme for making the data used in this study available.

## Author Contributions

**Conceptualization:** Waheed Babatunde Yahya.

**Data curation:** Lateef Babatunde Amusa, Waheed Babatunde Yahya.

**Formal analysis:** Lateef Babatunde Amusa.

**Methodology:** Lateef Babatunde Amusa.

**Supervision:** Waheed Babatunde Yahya.

**Writing – original draft:** Lateef Babatunde Amusa, Annah Vimbai Bengesai.

**Writing – review & editing:** Lateef Babatunde Amusa, Waheed Babatunde Yahya, Annah Vimbai Bengesai.

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
