## [Decision Letter · Decision Letter 0]

30 Jan 2023

PONE-D-22-26344Spatial variations and Determinants of malnutrition Among Under-five Children in Nigeria: A population-based cross-sectional studyPLOS ONE

Dear Dr. Amusa,

Thank you for submitting your manuscript to PLOS ONE. After careful consideration, we feel that it has merit but does not fully meet PLOS ONE’s publication criteria as it currently stands. Therefore, we invite you to submit a revised version of the manuscript that addresses the points raised during the review process.

Two experts in the field provide suggestions for improvement. In addition, and in order to expedite the review process and ensure completeness of reporting, PLOS ONE endorses the use of the STROBE checklist (http://www.strobe-statement.org). Please include the suggested checklist for observational studies noting where you are addressing the issues. Also ensure that statistical reporting guidelines are fulfilled (https://journals.plos.org/plosone/s/submission-guidelines.#loc-statistical-reporting). 

We look forward to receiving your revised manuscript.

Kind regards,

José Antonio Ortega, Ph.D.

Academic Editor

PLOS ONE

Journal Requirements:

4. We note that Figures 1, 3 and 5 in your submission contain [map/satellite] images which may be copyrighted. All PLOS content is published under the Creative Commons Attribution License (CC BY 4.0), which means that the manuscript, images, and Supporting Information files will be freely available online, and any third party is permitted to access, download, copy, distribute, and use these materials in any way, even commercially, with proper attribution. For these reasons, we cannot publish previously copyrighted maps or satellite images created using proprietary data, such as Google software (Google Maps, Street View, and Earth). For more information, see our copyright guidelines: http://journals.plos.org/plosone/s/licenses-and-copyright.

a. You may seek permission from the original copyright holder of Figures 1, 3 and 5 to publish the content specifically under the CC BY 4.0 license.  

Reviewers' comments:

Reviewer's Responses to Questions

**Comments to the Author**

1. Is the manuscript technically sound, and do the data support the conclusions?

Reviewer #1: Yes

Reviewer #2: Yes

2. Has the statistical analysis been performed appropriately and rigorously? 

Reviewer #1: Yes

Reviewer #2: Yes

3. Have the authors made all data underlying the findings in their manuscript fully available?

Reviewer #1: Yes

Reviewer #2: Yes

4. Is the manuscript presented in an intelligible fashion and written in standard English?

Reviewer #1: Yes

Reviewer #2: Yes

5. Review Comments to the Author

Reviewer #1: This paper has great potential and would be relevant to Nigeria if all amendments are given revised. The authors show great composure, especially in the methods and presentation of the results.

Material and methods:

Please use the following format for this section

Data management: [How to manage data]

Descriptive statistics: [How to summarize data]

Explorative statistics: [ inferential statistics you have used]

Results:

Please use the following format for this section

General characteristics: [status of the outcome variable, summarize the data]

Factors associated with outcome:

Reviewer #2: Thank you so much for requesting me to review this manuscript. The manuscript assessed the “Spatial variations and determinants of malnutrition among under five children in Nigeria: A population-based cross-sectional study”.

The manuscript is interesting. In general, it is well done but there are some comments that need to be fixed.

Introduction

1. Line 39: “ … from 6.6million …..2017” Reference(s) is needed.

Study variables

Table2. I wonder why you did not consider the following variables:

immunization card

History of immunization.

2. Line 93: Table1: I think you need to provide source and to test for the reliability of the index .

Specification of Bayesian prior distribution and hyper-parameters

3. Line159: “… and diffuse priors…..( you Wrote beta r1 or beta r1) and…this is confusing, is there any difference between them.

4. Did you consider possible interaction effect(s)?

5. I think mother’s nutrition knowledge may also be an important factor and it is not considered in this research. Why?

6. Line 334: … larger the family, the less per capita input” It is not clear please explain it.

7. Line 257: …48 moths, after which the effect decreased…” This interpretation is correct. Please observe the figure and interpret it correctly.

6. PLOS authors have the option to publish the peer review history of their article (what does this mean?). If published, this will include your full peer review and any attached files.

Reviewer #1: No

Reviewer #2: No

---

## [Author Response · Author response to Decision Letter 0]

20 Feb 2023

Journal Requirements:

Response: This is duly noted.

Response: Thank you. The minimal data set underlying the results described in our manuscript has been uploaded as a supporting information (SI) file.

Response: Ethics statement has now been moved to the Methods section of the manuscript.

4. We note that Figures 1, 3 and 5 in your submission contain [map/satellite] images which may be copyrighted. All PLOS content is published under the Creative Commons Attribution License (CC BY 4.0), which means that the manuscript, images, and Supporting Information files will be freely available online, and any third party is permitted to access, download, copy, distribute, and use these materials in any way, even commercially, with proper attribution. For these reasons, we cannot publish previously copyrighted maps or satellite images created using proprietary data, such as Google software (Google Maps, Street View, and Earth). For more information, see our copyright guidelines: http://journals.plos.org/plosone/s/licenses-and-copyright.

Response: Truly, Figure 1 may be copyrighted, and we have thus removed it from the submission. However, we can confirm that Figures 3 and 5 were created or constructed by the authors based on the study data as part of the spatial data analysis using R (version 4.2.1) package. Permission from a copyright holder is not applicable in this context. 

Reviewers’ comments

We would like to thank the reviewers for their comments as well as the opportunity to revise our manuscript for further consideration in your journal. We have addressed all the comments from the reviewers and below we provide a detailed description of the revisions made. In the revised manuscript, the green-shaded texts indicate changes made in response to suggestions and questions of the reviewers.

Reviewer #1

Comment: This paper has great potential and would be relevant to Nigeria if all amendments are given revised. The authors show great composure, especially in the methods and presentation of the results.

Material and methods:

Please use the following format for this section

Data management: [How to manage data]

Descriptive statistics: [How to summarize data]

Explorative statistics: [ inferential statistics you have used]

Results:

Please use the following format for this section

General characteristics: [status of the outcome variable, summarize the data]

Factors associated with outcome:

Response: The suggested presentation format is highly appreciated. We have implemented as suggested. However, your suggested format for the results section “General characteristics” has already been implied in the descriptive statistics section suggested under the material and methods section. 

Reviewer #2

Thank you so much for requesting me to review this manuscript. The manuscript assessed the “Spatial variations and determinants of malnutrition among under five children in Nigeria: A population-based cross-sectional study”.

The manuscript is interesting. In general, it is well done but there are some comments that need to be fixed.

Introduction

Comment: 1. Line 39: “ … from 6.6million …..2017” Reference(s) is needed.

Response: Thank you. The requested reference has now been cited.

Study variables

Comment: Table2. I wonder why you did not consider the following variables:

immunization card

History of immunization.

Response: We agree that these are important variables, and many more of such factors that potentially affect malnutrition; however, we are limited by the available variables in the NDHS database. Data on the suggested variables were not collected.

Comment: 2. Line 93: Table1: I think you need to provide source and to test for the reliability of the index.

Response: Though the CIAF index has been referenced in lines 57 and 58, we have now referenced the original source of the CIAF (Nandy & Svedberg, 2000) close to Table 1 in line 104. 

Specification of Bayesian prior distribution and hyper-parameters

Comment:3. Line159: “… and diffuse priors…..( you Wrote beta r1 or beta r1) and…this is confusing, is there any difference between them.

Response: The sentence is truly confusing. We have now rephrased (see lines 161,162).

Comment: 4. Did you consider possible interaction effect(s)?

Response: In the final analysis, no, we did not. In a preliminary analysis, the inclusion of interaction effects worsened our model fit, hence the need to drop them from the main analysis. 

Comment: 5. I think mother’s nutrition knowledge may also be an important factor and it is not considered in this research. Why?

Response: We agree that mother’s nutrition knowledge is a good potential predictor variable in this study; however, we are limited by the available variables in the NDHS database. Data on the suggested variable were not collected.

Comment: 6. Line 334: … larger the family, the less per capita input” It is not clear please explain it.

Response: Thank you, we have rephrased the sentence to read better (See lines 339, 340). We intended to imply the dilution model which predicts that larger families reduce the amount of resources (time, energy, money) available for each child, thus hindering social and physical development.

Comment: 7. Line 257: …48 moths, after which the effect decreased…” This interpretation is correct. Please observe the figure and interpret it correctly.

Response: Thanks for observing this interpretation error. We have now revised the interpretation.

Sincerely,

Lateef Amusa (PhD), (on behalf of all the authors)

---

## [Decision Letter · Decision Letter 1]

29 Mar 2023

Spatial variations and Determinants of malnutrition Among Under-five Children in Nigeria: A population-based cross-sectional study

PONE-D-22-26344R1

Dear Dr. Amusa,

We’re pleased to inform you that your manuscript has been judged scientifically suitable for publication and will be formally accepted for publication once it meets all outstanding technical requirements.

Kind regards,

José Antonio Ortega, Ph.D.

Academic Editor

PLOS ONE

Additional Editor Comments (optional):

Reviewers' comments:

Reviewer's Responses to Questions

**Comments to the Author**

1. If the authors have adequately addressed your comments raised in a previous round of review and you feel that this manuscript is now acceptable for publication, you may indicate that here to bypass the “Comments to the Author” section, enter your conflict of interest statement in the “Confidential to Editor” section, and submit your "Accept" recommendation.

Reviewer #1: All comments have been addressed

Reviewer #2: All comments have been addressed

2. Is the manuscript technically sound, and do the data support the conclusions?

Reviewer #1: Yes

Reviewer #2: Yes

3. Has the statistical analysis been performed appropriately and rigorously? 

Reviewer #1: Yes

Reviewer #2: Yes

4. Have the authors made all data underlying the findings in their manuscript fully available?

Reviewer #1: Yes

Reviewer #2: Yes

5. Is the manuscript presented in an intelligible fashion and written in standard English?

Reviewer #1: Yes

Reviewer #2: Yes

6. Review Comments to the Author

Reviewer #1: The authors have addressed all my comments. Therefore, based on my point of view, the research article is well written and informative.

Reviewer #2: (No Response)

7. PLOS authors have the option to publish the peer review history of their article (what does this mean?). If published, this will include your full peer review and any attached files.

Reviewer #1: No

Reviewer #2: No

---

## [Editor Report · Acceptance letter]

31 Mar 2023

PONE-D-22-26344R1 

Spatial variations and Determinants of malnutrition Among Under-five Children in Nigeria: A population-based cross-sectional study 

Dear Dr. Amusa:

I'm pleased to inform you that your manuscript has been deemed suitable for publication in PLOS ONE. Congratulations! Your manuscript is now with our production department. 

Kind regards, 

on behalf of

Dr. José Antonio Ortega 

Academic Editor

PLOS ONE